# Iron Chelation in Murine Models of Systemic Inflammation Induced by Gram-Positive and Gram-Negative Toxins

**DOI:** 10.3390/antibiotics9060283

**Published:** 2020-05-26

**Authors:** Danielle Fokam, Kayle Dickson, Kiyana Kamali, Bruce Holbein, Patricia Colp, Ashley Stueck, Juan Zhou, Christian Lehmann

**Affiliations:** 1Department of Physiology and Biophysics, Dalhousie University, Halifax, NS B3H 1X5, Canada; danielle.fokam@dal.ca; 2Department of Microbiology and Immunology, Dalhousie University, Halifax, NS B3H 1X5, Canada; kayle.dickson@dal.ca (K.D.); beholbein@sympatico.ca (B.H.); 3Department of Pharmacology, Dalhousie University, Halifax, NS B3H 1X5, Canada; kiyana.kamali@dal.ca; 4Department of Pathology, Dalhousie University, Halifax, NS B3H 1X5, Canada; p.colp@dal.ca (P.C.); ashley.stueck@dal.ca (A.S.); 5Department of Anesthesia, Pain and Perioperative Medicine, Dalhousie University, Halifax, NS B3H 1X5, Canada; juan.zhou@dal.ca

**Keywords:** inflammation, sepsis, iron chelation, intravital microscopy

## Abstract

Iron is an essential element for various physiological processes, but its levels must remain tightly regulated to avoid cellular damage. Similarly, iron plays a dual role in systemic inflammation, such as with sepsis. Leukocytes utilize iron to produce reactive oxygen species (ROS) to kill bacteria, but pathologically increased iron-catalyzed ROS production in sepsis can lead to damage of host cells, multi-organ failure and death. Temporary reduction in bioavailable iron represents a potential therapeutic target in sepsis. This study investigates the effect of the novel iron chelator, DIBI, in murine models of systemic (hyper-)inflammation: C57BL/6 mice were challenged with toxins from Gram-positive (*Staphylococcus aureus*: lipoteichoic acid, peptidoglycan) and Gram-negative bacteria (*Escherichia coli* and *Klebsiella pneumoniae*: lipopolysaccharide). Intravital microscopy (IVM) was performed to assess immune cell activation and its impact on microvascular blood flow in vivo in the microcirculation of the gut. Plasma inflammatory mediators were measured via multiplex assay, and morphologic change in intestinal tissue was evaluated. DIBI treatment decreased leukocyte (hyper-)activation induced by Gram-positive and Gram-negative toxins. In some cases, it preserved capillary perfusion, reduced plasma inflammatory markers and attenuated tissue damage. These findings support the utility of DIBI as a novel treatment for systemic inflammation, e.g., sepsis.

## 1. Introduction

Iron is one of the most abundant metal elements on earth [1]. This biomolecule is essential for the physiological processes of most forms of life, but its levels must remain tightly regulated to prevent damage. Iron serves as the catalyst for Fenton chemistry, which results in the production of highly reactive oxygen species (ROS) [2]. ROS are involved in pathogen clearance during the respiratory burst of phagocytic cells. However, during infection, iron plays a dual role since ROS overproduction can lead to oxidative stress and, ultimately, tissue damage [3].

Sepsis–the “worst case” of infection—is a global health concern because of its increasing incidence, high mortality, and the absence of an approved treatment for the underlying dysregulated immune response [4,5]. Essential basic treatments include surgery to control the source of infection, antibiotics, fluid resuscitation, vasopressors, blood transfusion, etc. [6]. During the critical hyper-inflammatory phase(s) of sepsis, iron acts as a catalyst to perpetuate the pro-inflammatory effects through ROS overproduction, e.g., increased leukocyte–endothelial interaction, reduced capillary blood flow and increased plasma cytokine levels. In addition, iron is a critical nutrient for bacterial growth. Properly reducing iron levels by iron chelation may provide an interesting therapeutic approach for sepsis—not only in the context of anti-oxidant therapies, which have proven to be advantageous in pre-clinical sepsis models but also in the context of inhibiting bacterial growth [7,8,9,10].

FDA-approved iron chelators include deferoxamine (DFO), deferiprone (DFP) and deferasirox (DFX). They are clinically used for iron overload, e.g., in sickle cell disease, hemochromatosis and thalassemia [11]. DIBI is a novel synthetic iron-binding polymer that is highly selective for iron, water-soluble and non-toxic in murine models [12,13,14] (Figure 1). It consists of a polyvinylpyrrolidone backbone containing nine 3-Hydroxy-1-(β-methacrylamidoethyl)-2-methyl-4(1*H*) pyridinone (MAHMP) residues per molecule [12]. The use of iron chelators in sepsis represents an innovative treatment approach [15,16,17,18]. Previous studies demonstrated that DIBI administration was able to reduce leukocyte activation and improve microcirculation in systemic inflammation induced by lipopolysaccharide from *E. coli*—a common preclinical model of sepsis [19,20]. However, these findings have not yet been validated in systemic inflammation induced by other bacterial toxins. Each toxin has a specific manner of triggering inflammatory pathways. While Gram-negative lipopolysaccharide (LPS) activates toll-like receptor 4 (TLR4), Gram-positive toxins activate mainly TLR2. Furthermore, species from the same group of bacteria can act differently [21]. The uniqueness of each bacterium (variation in amino acid sequence and other structural features) induces a specific immune response. As such, this study aims to assess potential anti-inflammatory effects of the novel iron chelator, DIBI, on the immune response to bacterial toxins from Gram-positive (lipoteichoic acid LTA and peptidoglycan PGN from *Staphylococcus aureus*) and Gram-negative (LPS from *Klebsiella pneumoniae* and *Escherichia coli*).

## 2. Results

### 2.1. IVM

#### 2.1.1. Leukocyte Adhesion

The primary outcome parameter used to assess systemic inflammation was the adhesion of leukocytes in both collecting and postcapillary venules of the intestinal submucosa. All toxins were able to induce inflammation by significantly increasing leukocyte adhesion (*p* < 0.05) in both collecting and postcapillary venules 2 h after toxin-challenge, with LPS_e_ the most potent Gram-negative toxin, while LTA was the most potent Gram-positive toxin. LPS from *K. pneumoniae* (LPS_k_) induced significantly less adhesion in collecting (−78% vs. LPS_e_) and postcapillary (−69% vs. LPS_e_) venules, respectively (Figure 2a–d). For the Gram-positive toxins, PGN induced significantly less adhesion than LTA in postcapillary venules (−36%; Figure 2d) but showed no significant difference in collecting venules. In collecting and postcapillary venules, the administration of DIBI significantly reduced leukocyte adhesion for all four toxins (LPS_e_, LPS_k_, LTA and PGN), but not back to control levels (Figure 2a–d). DIBI showed the greatest effect on PGN-induced leukocyte adhesion (75% reduction) and had the least effect on LPS_e_ (57% reduction) in collecting venules.

#### 2.1.2. Capillary Perfusion

Administration of Gram-negative toxins, LPS_e_ and LPS_k_, and Gram-positive toxins, LTA and PGN, induced a significant decrease (57%, 59%, 34% and 53%) of the functional capillary density (FCD) measured 2 h after induction of toxemia in the intestinal muscle layer compared to the control group (Figure 2e,f). A similar decrease was seen in the mucosa layer (Figure 2g,h). DIBI significantly improved the intestinal muscle layer FCD in the LPS_e_ + DIBI group (+45%) and PGN+DIBI group (+19%) but not in the LPS_k_ + DIBI and LTA + DIBI groups (Figure 2). In the intestinal mucosa layer, significant improvements in FCD by DIBI were only observed in the LPS_e_ + DIBI group (Figure 2g,h).

### 2.2. Cytokines

Three hours after toxin administration, a significant increase in TNF-α plasma levels compared to the control group (CON) was observed for all groups (LPS_e_, LPS_k_ and LTA) except animals that received PGN (Figure 3a,b). DIBI further increased TNF-α plasma levels significantly in animals that received LPS from *K. pneumoniae* (LPS_k_+DIBI), yet reduced the LTA-induced increases in TNF-α. DIBI did not have any significant effect on the TNF-α plasma level induced by LPS from *E. coli.*

In parallel to increased TNF-α plasma levels, there was a significant increase in IL-6 plasma levels for all groups (LPS_e_, LPS_k_ and LTA) except animals that received PGN (Figure 3c,d). Furthermore, the *E. coli* LPS challenge demonstrated significantly higher levels of IL-6 than the *K. pneumoniae* LPS challenge. The administration of DIBI significantly reduced IL-6 levels in the LTA+DIBI group (*p* < 0.05).

Three hours after toxin administration, a significant increase in IL-10 plasma levels relative to the control group (CON) was observed in all groups (Figure 3e,f). Moreover, the LTA challenge demonstrated significantly higher levels of IL-10 increase compared to PGN levels. DIBI administration significantly reduced IL-10 levels in the LTA+DIBI group (*p* < 0.05).

For IL-1β plasma levels, toxin administration only caused a significant increase in the *E. coli* LPS group (LPS_e_) compared to controls (Figure 3g,h). DIBI did not have a significant effect on that increase.

The toxin challenge significantly increased sICAM-1 plasma levels in all groups compared to the control group (Figure 3i,j). Treatment with DIBI significantly attenuated toxin-induced elevations of plasma sICAM-1 in all groups.

### 2.3. Histology

When assessing morphological changes in the intestine, the control group (CON) presented only minor morphological alterations (Chiu score [22]: grade 0–1). Gruenhagen′s spaces at the apex of the villi were observed intermittently (Figure 4b,c). Slightly more Gruenhagen′s spaces were seen at the apex of the villi in untreated Gram-positive toxins, yet there were no significant differences between untreated, DIBI-treated and control groups. However, DIBI significantly increased the Chiu score of the LTA+DIBI and PGN + DIBI-treated groups compared to the control (Figure 4d,e). For the Gram-negative toxins, only LPS from *E. coli* induced significant damage of the intestinal tissue 3 h after induction of inflammation compared to the control group (Figure 4d). For LPS_e_-challenged animals, there were more Gruenhagen′s spaces, with some of them extending to the subepithelial layer and inducing epithelial cells loss (grade 2 of Chiu score; Figure 4c). DIBI treatment significantly reduced the damage caused by LPS_e_. When comparing the two LPS groups, LPS_k_ produced significantly less histological damage than LPS_e_ (Figure 4d).

## 3. Discussion

The goal of this study was to assess the effects of the novel iron chelator, DIBI, on the immune response to systemic inflammation induced by Gram-positive and Gram-negative bacterial toxins. Since iron availability can potentially have different impacts on bacterial growth and the immune response and because microbial cell wall components, such as LPS that are mainly implicated in inflammatory responses, a sterile model of murine toxemia was chosen. This approach allowed us to hone in on the immune response associated with sepsis. Toxemia with an infection-free model involves the systemic administration of a controlled amount of a sterile bacterial toxin. These toxins are pathogen-associated molecular patterns (PAMPs), which trigger various inflammatory pathways. Inflammation was evaluated by intravital microscopy (IVM) of the intestine to measure leukocyte activation and capillary perfusion. The intestine plays a critical role during inflammation. There is a normal physiological balance between the host and its intestinal bacterial colonizers, which make up the gut microbiome. Sepsis pathophysiology, including tissue damage by oxidative stress and hypoxia, disrupts this homeostasis by damaging the intestinal mucosal barrier, producing a “leaky gut”. Intestinal flora is then more able to translocate across the gut into systemic circulation [23,24,25]. Under homeostatic conditions, intestinal flora is beneficial, but once it has leaked into the systemic circulation, it may contribute to overall pathogenesis and increase damage. In addition to leukocyte activation, inflammation and tissue damage were also evaluated by assessing plasma cytokine levels and histological assessment of intestinal tissue, respectively.

In our murine model of acute toxemia, administration of DIBI at a dosage of 10 mg/kg resulted in decreased leukocyte activation observed in the microcirculation by a reduction in the number of leukocytes adhering to the intestinal vascular endothelium and the decrease in sICAM-1 plasma levels.

In response to inflammation, leukocytes are typically attracted by endothelial cells, roll on the vascular endothelium, firmly adhere to the vessel wall and then transmigrate to the site of inflammation [26]. Quantifying leukocyte adhesion in collecting and postcapillary venules, using intestinal IVM, served as our primary experimental endpoint. Minimal baseline leukocyte adhesion was observed in control groups. Toxin doses used in our model of systemic inflammation were effective since all toxins significantly increased leukocyte adhesion in collecting and postcapillary venules when compared to control animals.

Although both Gram-positive and Gram-negative toxins have been shown to induce signs of systemic inflammation, few studies have compared toxins directly [27]. In this study, LPS from *E. coli* was more potent than LPS from *K. pneumoniae,* and LTA was more potent than PGN. According to the literature, LPS from *E. coli* tends to be more potent than LTA regarding leukocyte adhesion in vivo. When comparing the pro-inflammatory properties of different doses of LPS from *E. coli* and LTA from *S. aureus*, LPS was more potent than the same dose of LTA. Moreover, LTA did not necessarily demonstrate a dose-response in terms of leukocyte–endothelial interactions or cytokine secretion when administered at higher doses and assessed over a prolonged duration [28,29]. Yipp et al. [29] have shown in a murine model using IVM that intrascrotal administration of LTA (at doses similar, 10 or 100 times higher than LPS) induced 10 times less leukocytes adhesion in cremasteric postcapillary venules after 4 and 24 h observation than LPS (0.05 μg/kg). However, when LPS and LTA were administered systemically at doses comparable to our experiments (5 mg/kg), there was no significant difference between LPS and LTA on leukocyte adhesion in cremasteric postcapillary venules. Finney et al. found similar results when comparing *S. aureus* LTA to *Salmonella enteritidis* LPS [30]. Our results for *K. pneumoniae* LPS-induced inflammation were surprising. Clinical studies have suggested that *K. pneumoniae* bacteremia may lead to worsened outcomes when compared to *E. coli*. [31]. The difference between that study and ours may be attributed to the fact that the patients had *K. pneumoniae* infection (whole bacteria, not just toxins). Toxic products from bacterial growth, as well as tissue damages due to bacterial destruction, may have made *K. pneumoniae* more lethal than *E. coli*. Therefore, more studies are needed to validate our findings that suggest LPS from *K. pneumoniae* is less potent than both Gram-positive toxins and LPS from *E. coli* in terms of inflammatory activation. The fact that the PAMPs from Gram-positive and Gram-negative bacteria trigger different inflammatory pathways might also explain the differences in the immune response to the LPS toxins of those species. Gram-negative LPS primarily activates TLR4, while Gram-positive LTA and PG act on TLR2 [21]. Variability in potency can also occur even when molecules trigger the same signaling pathway, as seen when comparing LPS from *E. coli* and *K. pneumoniae*, or LTA and PGN from *S. aureus*. Changes in amino acid sequence and other structural features give each PAMP a unique ability to induce a specific immune response [21].

According to the literature, PAMPs increase the expression of adhesion molecules and increase leukocyte–endothelial cell interaction through a process involving reactive oxygen species (ROS) and iron [2]. As previously observed, our novel iron chelator, DIBI, was able to significantly reduce the toxin-increased adhesion of leukocytes for all toxins in collecting and postcapillary venules [19]. Leukocyte adhesion was not reversed back to baseline levels, indicating a decrease in over-activation and the preservation of an appropriate inflammatory response. These results agree with findings from Arora et al., which showed anti-inflammatory effects when using DIBI in LPS-induced rodent ocular inflammation. DIBI-treated animals showed reduced ocular inflammation in local and systemic induced uveitis [20]. Similar anti-inflammatory effects have also been observed for other iron chelators. Messaris et al. noted a survival benefit in septic rats treated with deferoxamine, which may be attributed to a reduction in oxidative damage [18]. When considering the results of this study in conjunction with previous in vivo and in vitro studies, there is strong evidence for the anti-inflammatory effects of iron chelation.

Microcirculatory dysfunction represents a major issue in the pathophysiological cascade of systemic inflammation and is an independent predictor of mortality in sepsis [32,33]. IVM was used to measure capillary blood flow in intestinal muscle and mucosa layers. The mucosa layer receives 80% of the total intramural blood flow, and as such, may be more sensitive to damages when compared to the muscle layer [34]. As expected, all toxins significantly decreased functional capillary density in intestinal muscle and mucosa layers when compared to control. The effect of DIBI on the changes in FCD changes was variable. DIBI improved capillary density when administered to animals with toxemia induced by LPS from *E. coli* and PGN from *S. aureus.* No significant improvements were seen in the case of LPS from *K. pneumoniae* or LTA from *S. aureus*. Other studies corroborate the effects of DIBI in similar *E. coli* LPS-induced toxemia models. Findings from Thorburn et al. using IVM showed that DIBI restored the LPS-decreased intestinal capillary blood flow back to control levels in endotoxemic mice [35]. Arora et al. reported that DIBI attenuated the LPS induced reduction in iridial microvascular perfusion in experimental uveitis in both mice and rats [20]. To understand why DIBI was not able to restore perfusion in the case of LPS from *K. pneumoniae* and LTA from *S. aureus*, it is important to consider the multifactorial nature of changes in capillary blood flow. Factors, such as capillary leakage, disseminated intravascular coagulation, increased leukocyte–endothelial interaction, endothelial cell dysfunction, systemic redistribution of blood flow and tissue edema, affect capillary blood flow in this context [34,36]. These parameters may vary with the administration of different toxins, masking the impact of DIBI on microvascular perfusion.

Quantification of mediators of inflammation allowed for further assessment of the activation of inflammatory pathways in the toxemia model. Plasma levels of inflammatory cytokines and soluble adhesion molecules were measured 3 h after toxin administration. Cytokines act either pro- or anti-inflammatory molecules and are released by several cells, e.g., immune cells and endothelial cells [37,38]. Cytokine levels vary according to the intensity of inflammation, and patterns of expression may help predict mortality in sepsis [39]. In our study, LPS from *E. coli* was the only toxin that caused a significant increase across all inflammatory parameters, mimicking the cytokine storm often observed in sepsis [40]. With respect to the other toxins administered, higher variability was observed in terms of cytokine expression. This may be related to differences in the inflammatory pathways activated or structural differences between PAMPs. Comparable results were seen by others when measuring plasma TNF-α, IL-6 and IL-10 levels in various organs following the *S. aureus* PGN challenge. Wang et al. observed elevated levels of cytokines in both plasma and organs following a PGN challenge, which exceeded levels seen in our toxemia model [41]. Since the authors used a dose of 10 mg/kg of PGN, this increase in cytokine expression may indicate a possible dose-response effect. Another study using *S. aureus* LTA to stimulate human dermal, microvascular endothelial cells showed increased IL-6 production [42]. In contrast to this, a similar experiment on J774 cells using *S. aureus* LTA showed extremely low cytokine levels [28]. In a context of sepsis-like inflammation, where high levels of cytokine are expected, these findings could be explained by differences in study design, such as differences in doses or species. Alternatively, the mixed nature of the inflammatory response in sepsis, where both hyper-inflammation and immunosuppression may occur, can explain such discrepancies.

DIBI administration was only able to reduce cytokine plasma levels in some cases. Notably, DIBI treatment significantly reduced cytokine levels for animals challenged with LTA across all parameters. This represents a novel finding, as DIBI was previously only evaluated in the context of LPS from *E. coli*. A recent study using the iron chelator, di-2-pyridylketone-4,4-dimethyl-3-thiosemicarbazone, identified a dose-dependent reduction in serum levels of IL-1β and TNF-α in a murine model of allergic rhinitis [43]. Additional studies using iron chelation in endotoxemia models also showed a decrease in LPS-induced activation of NF-κB and reduced plasma levels of TNF-α, IL-6, IL-1β and IL-10 [16,17,44,45]. Our findings showed a TNF-α plasma level increase in DIBI-treated animals challenged with LPS from either *E. coli* or *K. pneumoniae*. Toxins appear to induce different patterns of cytokine release in experimental models of sepsis depending on toxin dosage, route of administration and model used. Although detrimental side effects from DIBI administration cannot be completely ruled out, an increase in cytokine plasma levels measured at only one timepoint might not reflect a “real” increase in cytokine expression (Figure 5). Alternatively, a shift in the course of cytokine release (delay of cytokine peak—dark blue curve) or a combined decrease and delay of the cytokine release (light blue curve) can explain the unexpected data. Future investigation of cytokine levels at various time points is needed.

Morphological intestinal tissue damage was assessed in our study using the Chiu scoring system [22]. Only mice challenged with LPS from *E. coli* showed a significant increase in tissue damage when compared to the control group. The observed damage was still in the lower range of the Chiu score suggesting minimal damage occurring at the early time point used in our study. DIBI administration was able to significantly reduce those minimal *E. coli* LPS-induced changes. This result was corroborated by results of Hu et al. [46] who saw in a study investigating the role of interferons therapy in murine polybacterial, abdominal sepsis minimal histological changes (maximum damages equivalent to grade 3 of Chiu score) 5 h post sepsis induction. Islam et al. also observed similar minimal histological changes 16 h post polybacterial sepsis in mice [47]. Based on these observations, it appears that much longer endotoxemia would be required to observe clinically relevant intestinal morphological changes due to systemic inflammation and to assess the impact of iron chelation.

Our study has several limitations, including the fact that the experiments were conducted in mice, which limits the applicability of our findings to humans, despite genetic and physiological similarities between humans and mice [48]. Furthermore, experiments were conducted using young and healthy male mice. This specific population cannot be considered representative of septic patients, which often are elderly and show co-morbidities. In our model, mice received a single challenge with a controlled amount of toxin, which does not mimic the clinical course of the disease. This model is limited in its representation of sepsis pathophysiology as it only examines hyperinflammation. Future research should use a more clinically relevant model of sepsis, where an infection is present. Despite improvements in inflammatory parameters, the impact of DIBI administration on functional outcomes and survival in sepsis has yet to be determined.

## 4. Materials and Methods

### 4.1. Bacterial Toxins

All toxins were purchased from Sigma–Aldrich (Oakville, ON, Canada) and used without further purification, as indicated by the supplier. Gram-negative toxins included LPS from *Escherichia coli* (LPS_e_; O26:B6) and from *Klebsiella pneumoniae* (LPS_k_; L4268). Lipoteichoic acid (LTA, L2515) and peptidoglycan (PGN, 77140) from *Staphylococcus aureus* were used as Gram-positive toxins. The novel iron chelator, DIBI, was provided by Chelation Partners Inc., Halifax, NS, Canada.

### 4.2. Animals

This study was carried out on eight to ten-week-old male C57BL/6 mice (Charles River Laboratories International Inc, Saint-Constant, QC, Canada) housed in the Carleton Animal Care Facility (Faculty of Medicine, Dalhousie University, Halifax, NS, Canada). Animals received a standard diet and water ad libitum and were maintained on a 12 h light/dark cycle in a 22 °C room. Weights ranged from 20 to 30 g. All experiments were approved by the University Committee on Laboratory Animals at Dalhousie University under protocol number #18-057 and were performed following the guidelines and standards of the Canadian Council on Animal Care.

### 4.3. Experimental Groups

Nine groups of animals (*n* = 5–6/group) were established for the study. Systemic inflammation was induced by 5mg/kg of the toxin with groups including LPS_e_, LPS_k_, LTA or PGN. LPS_e_ + DIBI, LPS_k_ + DIBI, LTA + DIBI, PGN + DIBI groups were treated with 10 mg/kg of DIBI. Control animals received the same weight-adjusted amount of vehicle (normal saline). Pilot experiments had shown no impact of DIBI alone treatment in healthy animals, so DIBI treatment only groups were not included in this study.

### 4.4. Experimental Model

Animals were anesthetized by an intraperitoneal (i.p.) injection of sodium pentobarbital (90 mg/kg). Depth of anesthesia was assessed by pedal reflex and maintained using sodium pentobarbital bolus administration (9 mg/kg). Mice were maintained at a body temperature of 37 ± 0.5 ℃ and received supplemental oxygen if irregular breathing occurred. Surgery for vascular access was performed, as previously described by Thorburn et al. [19]. In brief, the anterior-lateral neck area was shaved and disinfected using isopropyl alcohol. A small incision was made over the right jugular vein, which was further exposed using blunt dissection. The vessel was then tied off at the cranial site. At the caudal site of the vessel, a small incision was made to allow the insertion of a custom-made polyethylene catheter (PE 10), which was then secured using string. Toxins were administered intravenously (i.v.) through the catheter. After 15 min, intravenous treatment with DIBI/vehicle was administered.

### 4.5. Intravital Microscopy

Intravital microscopy (IVM) was performed 120 min after toxin challenge. Fluorescent dyes (1.5 mL/kg 0.05% Rhodamine 6 G; 1 mL/kg 5% fluorescein isothiocyanate albumin; Sigma–Aldrich, Oakville, ON, Canada) were administered i.v. 15 min prior to IVM. IVM was performed on the terminal ileum, as described previously [47,49]. In brief, the abdomen was opened after the injection of fluorescent dyes, and a loop of the terminal ileum was placed on a customized IVM stage [50,51]. Leukocyte adhesion was evaluated in both, collecting (V1) and postcapillary (V3) venules of the intestinal submucosa. Functional capillary density (FCD) was measured in the intestinal muscle layers and mucosal villi and was quantified by measuring the total length of perfused capillaries in a defined area. Six videos of 30 s were obtained for each parameter and then analyzed offline using ImageJ (National Institute of Health, Bethesda, MD, USA). Following IVM, blood samples were drawn by cardiac puncture, and animals were sacrificed by cervical dislocation.

### 4.6. Plasma Cytokine Measurements

Blood samples were centrifuged at 3000 rpm for 10 min to obtain plasma. Plasma levels of selected inflammatory cytokines (TNFα, IL-6, IL-1β, IL-10 and soluble ICAM-1) were analyzed according to manufacturer protocols using a Procarta Multiplex Cytokine Assay kit (Affymetrix; Freemont, CA, USA) and a Bio-Plex instrument with Bio-Plex software (Bio-Rad, Mississauga, ON, Canada).

### 4.7. Histology

Samples from the terminal ileum not affected by IVM were collected following euthanasia. Samples were fixed in 10% phosphate-buffered formalin (Fisher Scientific, Shanghai, China) for 24 h and processed using a Leica Pearl Wax Tissue Processor (Leica Microsystems, Wetzlar, Germany). Processed tissues were then sectioned (5 mm), and Hematoxylin and Eosin (H&E) stain performed. Tissues were assessed for histological damage using the scoring system developed by Chiu et al. [22]: grade 0, normal histology; grade 1, development of subepithelial Gruenhagen′s spaces, usually at the villous apex; grade 2, extension of Gruenhagen′s spaces with the moderate lifting of the epithelial layer; grade 3, massive epithelial lifting down the sides of villi +/− denudation of epithelium at the villous tip; grade 4, denuded epithelium with lamina propria and dilated capillaries exposed, possible increase in lamina propria cellularity by inflammatory cells; grade 5, digestion and disintegration of lamina propria with hemorrhage and ulceration.

### 4.8. Statistical Analysis

All data were analyzed using the statistical software package Prism 8 (version 8.2.0 272; GraphPad Software, La Jolla, CA, USA). Normal distribution was tested by the Kolmogorov–Smirnov test. One-way ANOVA followed by Newman Keuls’ test for multiple comparisons was used to analyze normally distributed data and post-hoc Kruskal–Wallis analysis for data not normally distributed. Differences at *p* < 0.05 were considered statistically significant.

## 5. Conclusions

The present study investigated the impact of the novel iron chelator, DIBI, on systemic inflammation in vivo induced by different Gram-negative and Gram-positive toxins. DIBI had previously been validated using an endotoxemia model, where LPS from *E. coli* served as the inflammatory trigger [19]. This study expanded upon these findings to demonstrate DIBI’s efficacy for other Gram-negative and Gram-positive toxins. While each toxin had different potency, DIBI demonstrated anti-inflammatory activity against each toxin by reducing leukocyte (over-)activation. DIBI also showed other anti-inflammatory effects, including improved capillary perfusion, reduced plasma cytokine levels and attenuated intestinal tissue damage depending on the chosen toxin. Further research is required to understand the variations in DIBI’s efficacy in relation to various PAMPs. These results provide further evidence supporting a role for iron chelation in treating (systemic) hyperinflammation. DIBI treatment represents a promising adjunct therapy for sepsis caused by Gram-positive and Gram-negative bacteria.

## Figures and Tables

**Figure 1 antibiotics-09-00283-f001:**
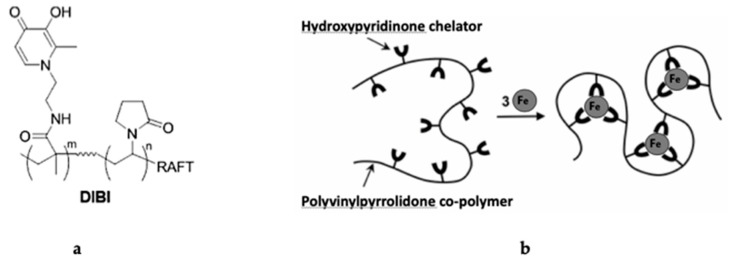
Structure of the DIBI polymer: (**a**) chemical structure of DIBI; nine 3-Hydroxy-1-(β-methacrylamidoethyl)-2-methyl-4(1*H*)-pyridinone (MAHMP) residues on a Polyvinylpyrrolidone (PVP) backbone; (**b**) representation of iron molecules arrangement by DIBI [14]–Reproduced by permission of The Royal Society of Chemistry.

**Figure 2 antibiotics-09-00283-f002:**
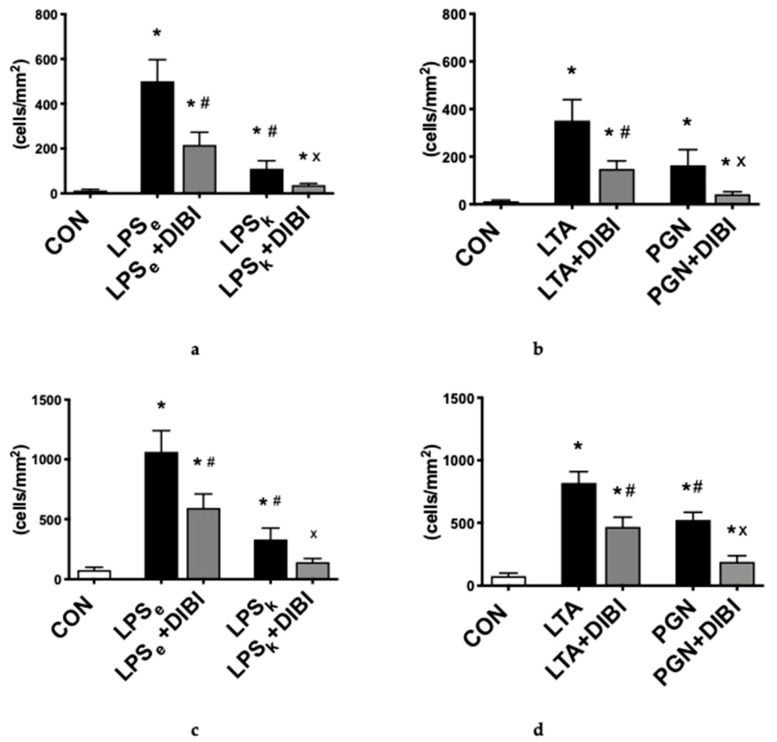
Intestinal intravital microscopy showing DIBI effects 2 h after challenge with 5 mg/kg of toxins from Gram-negative and Gram-positive bacteria: leukocyte adhesion in intestinal collecting venules after challenge with Gram-negative bacteria (**a**) and Gram-positive bacteria (**b**). Leukocyte adhesion in intestinal postcapillary venules after challenge with Gram-negative (**c**) and Gram-positive bacteria (**d**); Firmly adherent leukocytes per area of submucosal venules (<100μm vessel diameter) are quantified as cells/mm^2^. Functional capillary density (FCD) changes in the intestinal muscle layer after challenge with Gram-negative (**e**) and Gram-positive bacteria (**f**). FCD change in the intestinal mucosa layer after challenge with Gram-negative (**g**) and Gram-positive bacteria (**h**); FCD is calculated as the total length of perfused capillaries within a predetermined field and is quantified as cm/cm^2^. Each bar graph represents mean values ± SEM (n = 5–10 per group). The white bar represents animals from the control group, the black bar represents untreated toxemia groups, and the gray bars represent treated toxemia groups. * *p* < 0.05 compared to CON group; ^#^
*p* < 0.05 compared to LPS_e_/LTA group; ^x^
*p* < 0.05 compared to LPS_k_/PGN group.

**Figure 3 antibiotics-09-00283-f003:**
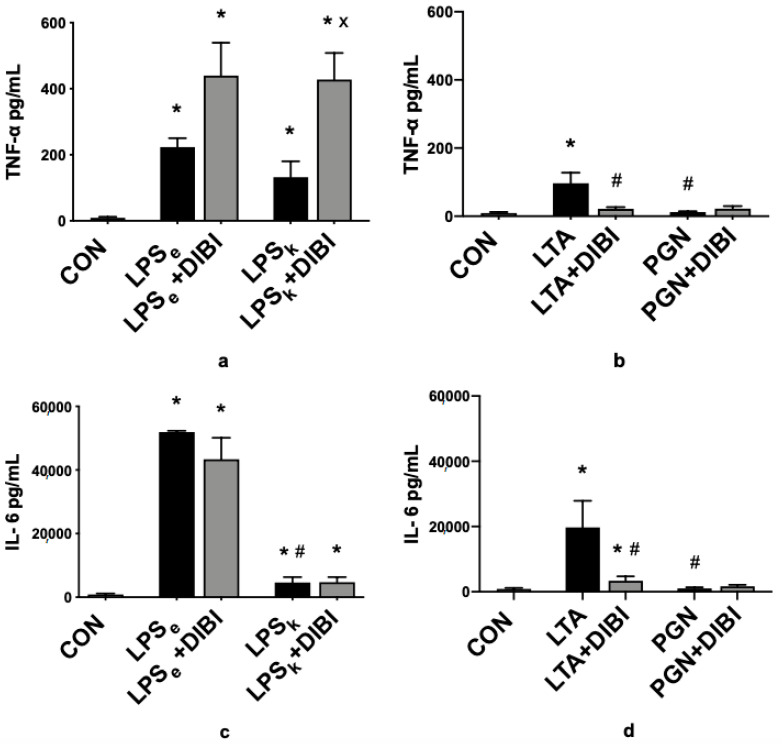
DIBI effects on plasma cytokines 2 h after challenge with 5 mg/kg of toxins (pg/mL): TNF-α plasma levels from Gram-negative (**a**) and Gram-positive (**b**) groups; IL-6 plasma levels from Gram-negative (**c**) and Gram-positive (**d**) groups; IL-10 plasma levels from Gram-negative (**e**) and Gram-positive (**f**) groups; IL-1β plasma levels from Gram-negative (**g**) and Gram-positive (**h**) groups; sICAM-1 plasma levels from Gram-negative (**i**) and Gram-positive (**j**) groups. Each bar graph represents mean values ± SEM (*n* = 5–12 per group). The white bar represents animals from the control group, the black bar represents untreated toxemia groups, and the gray bars represent treated toxemia groups. * *p* < 0.05 compared to CON group; ^#^
*p* < 0.05 compared to LPS_e_/LTA group; ^x^
*p* < 0.05 compared to LPS_k_/PGN group.

**Figure 4 antibiotics-09-00283-f004:**
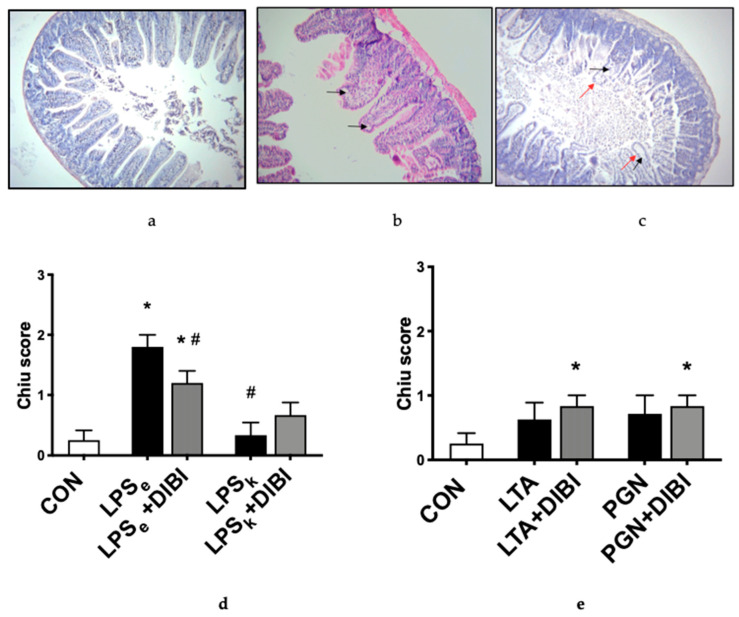
Chiu score [22] of a murine intestinal cross-section showing different degrees of tissue damage 3 h after toxin challenge. Hematoxylin and Eosin (H&E) staining under 40× magnification. (**a**) grade 0: Normal mucosal villi, epithelial cells well aligned and intact; (**b**) grade 1: Development of spaces called Gruenhagen′s space at the apex of villi (shown by black arrow), epithelial cells well aligned and intact; (**c**) grade 2: Extension of Gruenhagen′s space along the villi with damaged/missing epithelial cells shown by red arrow; (**d**) DIBI effects on morphological intestinal changes caused by Gram-negative and (**e**) Gram-positive toxins. Each bar represents mean values ± SEM. The white bars represent control groups, the black bars represent toxins groups (LPS_e_ from *E. coli*; LPS_k_ from *K. pneumoniae*; LTA and PGN from *S. aureus*), and the gray bars represent DIBI treated groups. * *p* < 0.05 compared to CON group; ^#^
*p* < 0.05 compared to LPS_e_/LTA group; ^x^
*p* < 0.05 compared to LPS_k_/PGN group.

**Figure 5 antibiotics-09-00283-f005:**
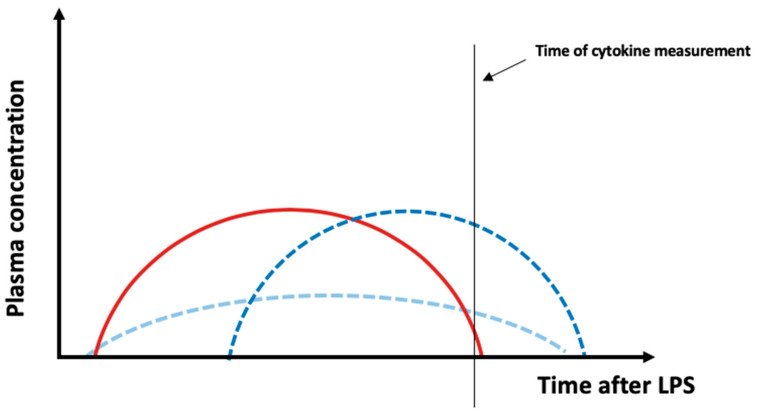
Possible effects of DIBI on plasma cytokine levels. In this study, plasma inflammatory parameters were measured at one time point. Assuming that the red curve represents the time course of a cytokine release after LPS administration, it can be speculated that an increase in plasma cytokine levels due to DIBI measured at only one timepoint might not be a “real” increase in cytokine expression but a right shift, i.e., delayed peak—dark blue curve) or a combination of a delayed peak and a reduction in the peak (light blue curve). In the two mentioned cases, there will be the same total plasma concentration, yet different curve shapes.

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
