# Peer review of "Iron Chelation in Murine Models of Systemic Inflammation Induced by Gram-Positive and Gram-Negative Toxins"

_antibiotics, 2020, doi:10.3390/antibiotics9060283_

Round 1
Reviewer 1 Report
The manuscript "Iron Chelation in Murine Models of Systemic Inflammation Induced by Gram-positive and Gram-negative Toxins" is presented in a very clear and comprehensive way. The study is very interesting and, despite the fact that further work needs to be done in the field, it gives a very good insight on the action (and potential) of the new siderophore.
Just notice some minor mistakes like "Figure 4" on line 112 should be "Figure 2".
Author Response
RESPONSE TO REVIEWERS 1
Comments to the Author: The manuscript "Iron Chelation in Murine Models of Systemic Inflammation Induced by Gram-positive and Gram-negative Toxins" is presented in a very clear and comprehensive way. The study is very interesting and, despite the fact that further work needs to be done in the field, it gives a very good insight on the action (and potential) of the new siderophore.Just notice some minor mistakes like "Figure 4" on line 112 should be "Figure 2".
Response: Thank you very much for this encouraging comment! The figure number has been revised in the manuscript.
Reviewer 2 Report
The mansucript is quite well written.
The authors demonstrated that DIBI treatment decreased leukocyte hyperactivation induced by Gram-positive and Gram-negative toxins. In some cases, it preserved capillary perfusion, reduced plasma inflammatory markers and attenuated tissue damage. These findings support the utility of DIBI as a novel treatment for systemic inflammation, such as sepsis.
The methods are adequate. The results justify the conclusions drawn. It would be useful for the readers to include the discussion of PMID: 28074932.
Author Response
RESPONSE TO REVIEWER 2
- Comments to the Author: The manuscript is quite well written. The authors demonstrated that DIBI treatment decreased leukocyte hyperactivation induced by Gram-positive and Gram-negative toxins. In some cases, it preserved capillary perfusion, reduced plasma inflammatory markers and attenuated tissue damage. These findings support the utility of DIBI as a novel treatment for systemic inflammation, such as sepsis.
Response: Thank you very much for this positive comment!
- Comments to the Author: The methods are adequate. The results justify the conclusions drawn. It would be useful for the readers to include the discussion of PMID: 28074932.
Response: Thank you again for your comment. According to the PMID you gave us, we found a paper named “The Helicobacter cinaedi antigen CAIP participates in atherosclerotic inflammation by promoting the differentiation of macrophages in foam cells” (https://www.ncbi.nlm.nih.gov/pmc/articles/PMC5225449/). Perhaps there was an error in the reference, and we saw the wrong article because we could not find a direct link between that paper and ours.
Reviewer 3 Report
This manuscript by Fokam et al describes the results of a controlled study assessing the ability of an iron chelator (DIBI) to mitigate the inflammatory impact of bacterial toxins in a murine model. Inflammatory biomarkers are measured as well as histology scores for vascular leakage. The authors conclude based on their results that DIBI could potentially alleviate some of the symptoms of systematic inflammation from sepsis.
The strength of this paper is that multiple measures are used to access inflammation and multiple inflammatory molecules provide a rough idea about the generality of the effect. The principle weakness is that a great deal has already been published describing the use of chelating agents in combating systematic inflammation (see Vulcano et al International Journal of Immunopharmacology, 31 Jul 2000, 22(8):635-644 or Srinivasan et al J. Immunology 2012 189:1911-1919). In fact, this same group published a very similar study two years ago also using DIBI and systematic LPS in a murine model (DOI: 10.3233/CH-189109). This group published another paper four years ago, with what I would say is a somewhat more clinically-relevant model, reaching exactly the same conclusion as this one, that DIBI was potentially useful in treating sepsis (https://www.ncbi.nlm.nih.gov/pubmed/26235905). Without a more mechanistic approach, which might explain why DIBI is more effective in some instances that others, I struggle to see what this paper adds to what is already in the literature.
Other comments:
-The text in Figures 2 and 3 was barely legible in the pdf I reviewed, the 'e' in LPSe was basically just a circle.
-In these same figures, significance is hard to follow because it isn't marked between groups (even when noted in the text), only being shown compared to the control group. Significance should be marked between groups and the p-values for those comparisons shown.
-Y axis titles would also improve the readability of these figures (ex. ("pg/ml TNF-alpha", "pg/ml IL-6", etc instead of simply "pg/ml")
-In the cytokine results, I can't reconcile the two bolded statements below:
2.2 Cytokines
Three hours after toxin administration, a significant increase of TNF-α plasma levels compared to the control group (CON) was observed for all groups (LPSe, LPSk and LTA) except animals that received PGN (figure 3 a and b). DIBI further increased TNF-α plasma levels significantly in animals that received LPS from K. pneumoniae (LPSk+DIBI) yet reduced the LTA-induced increases of TNF-α. No significant changes were observed in the LPSe group.
- Figure 1b is an exact copy of a figure from this paper (https://pubs.rsc.org/en/content/articlelanding/2018/md/c8md00192h#!divAbstract) for which I could find no attribution in this manuscript (e.g. Figure 1B is adapted from ...). There is an overlapping author and perhaps it's his figure, but it still cannot be used without at least acknowledging it's prior publication (and most journals still require permission to be sought).
Author Response
RESPONSE TO REVIEWER 3
- Comments to the Author: This manuscript by Fokam et al describes the results of a controlled study assessing the ability of an iron chelator (DIBI) to mitigate the inflammatory impact of bacterial toxins in a murine model. Inflammatory biomarkers are measured as well as histology scores for vascular leakage. The authors conclude based on their results that DIBI could potentially alleviate some of the symptoms of systematic inflammation from sepsis. The strength of this paper is that multiple measures are used to access inflammation and multiple inflammatory molecules provide a rough idea about the generality of the effect.
Response: Thank you very much for the time you took to review our paper and for all your helpful comments. Thank you also for recognizing the strengths of our manuscript.
- Comments to the Author: The principle weakness is that a great deal has already been published describing the use of chelating agents in combating systematic inflammation (see Vulcano et al International Journal of Immunopharmacology, 31 Jul 2000, 22(8):635-644 or Srinivasan et al J. Immunology 2012 189:1911-1919). In fact, this same group published a very similar study two years ago also using DIBI and systematic LPS in a murine model (DOI: 10.3233/CH-189109). This group published another paper four years ago, with what I would say is a somewhat more clinically-relevant model, reaching exactly the same conclusion as this one, that DIBI was potentially useful in treating sepsis (https://www.ncbi.nlm.nih.gov/pubmed/26235905). Without a more mechanistic approach, which might explain why DIBI is more effective in some instances that others, I struggle to see what this paper adds to what is already in the literature.
Response: Thank you very much for this important comment. As you mentioned, some studies have already evaluated the anti-inflammatory properties of iron chelating agents such as DIBI and are referenced and discussed in this manuscript (including the papers you cited). Nevertheless, they were predominantly done using LPS from E. coli as a toxin. However, different toxins trigger different inflammatory pathways: Gram-negative toxins activate primarily Toll-Like Receptors 4 (TLR4) while Gram-positive toxins activate mainly TLR2.(1)(2)(3). Therefore, it is useful to study the effects of iron chelators such as DIBI using different toxins. The following sentences have been added to the introduction:
“While Gram-negative LPS activates Toll-Like Receptor 4 (TLR4), Gram-positive toxins activate mainly TLR2. Furthermore, species from the same group of bacteria can act differently [35]. The uniqueness of each bacterium (variation in amino acid sequence and other structural features) induces a specific immune response.”
- Ramachandran G. Gram-positive and gram-negative bacterial toxins in sepsis: a brief review. Virulence [Internet]. 2014 Jan 1 [cited 2019 Apr 26];5(1):213–8. Available from: http://www.ncbi.nlm.nih.gov/pubmed/24193365
- Paul-Clark MJ, Mc Master SK, Belcher E, Sorrentino R, Anandarajah J, Fleet M, et al. Differential effects of Gram-positive versus Gram-negative bacteria on NOSII and TNFα in macrophages: Role of TLRs in synergy between the two. Br J Pharmacol. 2006 Aug 24;148(8):1067–75.
- Dickson K, Lehmann C. Inflammatory response to different toxins in experimental sepsis models. Vol. 20, International Journal of Molecular Sciences. MDPI AG; 2019.
- Comments to the Author: The text in Figures 2 and 3 was barely legible in the pdf I reviewed, the 'e' in LPSe was basically just a circle. In these same figures, significance is hard to follow because it isn't marked between groups (even when noted in the text), only being shown compared to the control group. Significance should be marked between groups and the p-values for those comparisons shown. Y axis titles would also improve the readability of these figures (ex. ("pg/ml TNF-alpha", "pg/ml IL-6", etc instead of simply "pg/ml")
Response: Thank you for your input. Revisions have been made to make the figures clearer and more visible.
- Comments to the Author: In the cytokine results, I can't reconcile the two bolded statements below:
2.2 Cytokines
Three hours after toxin administration, a significant increase of TNF-α plasma levels compared to the control group (CON) was observed for all groups (LPSe, LPSk and LTA) except animals that received PGN (figure 3 a and b). DIBI further increased TNF-α plasma levels significantly in animals that received LPS from K. pneumoniae (LPSk+DIBI) yet reduced the LTA-induced increases of TNF-α. No significant changes were observed in the LPSe group.
Response: Thank you for your comment. We are sorry for the confusion. The last sentence has been replaced in the manuscript by the following:
“DIBI did not have any significant effect on the TNF-α plasma level induced by LPS from E. coli.”
- Comments to the Author: Figure 1b is an exact copy of a figure from this paper (https://pubs.rsc.org/en/content/articlelanding/2018/md/c8md00192h#!divAbstract) for which I could find no attribution in this manuscript (e.g. Figure 1B is adapted from ...). There is an overlapping author and perhaps it's his figure, but it still cannot be used without at least acknowledging it prior publication (and most journals still require permission to be sought).
Response: Thank you for pointing out our error of omission. The description of the figure 1 has been replaced in the manuscript by the following:
“Figure 1. Structure of DIBI polymer: (a) chemical structure of DIBI; nine 3-hydroxy-1-(β-methacrylamidoethyl)-2-methyl-4(1 H)-pyridinone (MAHMP) residues on a PVP backbone; (b) representation of iron molecules arrangement by DIBI [14] – Reproduced by permission of The Royal Society of Chemistry.”
- The manuscript has been edited by native speakers.
Round 2
Reviewer 3 Report
I think the improvements to the figures have improved the readability of this paper. I hope that the field is able to make use of it.